# RETRACTED: Pneumonia in Patients with Chronic Lymphocytic Leukemia Treated with Venetoclax-Based Regimens: A Real-World Analysis of the Polish Adult Leukemia Group (PALG)

**DOI:** 10.3390/cancers16244168

**Published:** 2024-12-13

**Authors:** Elżbieta Kalicińska, Paula Jabłonowska-Babij, Marta Morawska, Elżbieta Iskierka-Jażdżewska, Joanna Drozd-Sokołowska, Ewa Paszkiewicz-Kozik, Łukasz Szukalski, Judyta Strzała, Urszula Gosik, Jakub Dębski, Iga Andrasiak, Anna Skotny, Krzysztof Jamroziak, Tomasz Wróbel

**Affiliations:** 1Clinical Department of Hematology, Cell Therapies and Internal Diseases, Wroclaw Medical University, 50-367 Wroclaw, Poland; pjablonowska@usk.wroc.pl (P.J.-B.); tomasz.wrobel@umw.edu.pl (T.W.); 2Experimental Hematooncology Department, Medical University of Lublin, 20-059 Lublin, Poland; mmorawska79@gmail.com; 3Department of General Hematology, Copernicus Memorial Hospital, Medical University of Lodz, 90-419 Lodz, Poland; elzbieta.iskierka-jazdzewska@umed.lodz.pl; 4Department of Hematology, Transplantation and Internal Medicine, Medical University of Warsaw, 02-091 Warsaw, Poland; johna.dr@poczta.fm (J.D.-S.); k.m.jamroziak@gmail.com (K.J.); 5Department of Lymphoid Malignancies, National Research Institute of Oncology, 02-781 Warsaw, Poland; ewa.paszkiewicz-kozik@pib-nio.pl; 6Department of Hematology, Collegium Medicum in Bydgoszcz, Nicolaus Copernicus University in Toruń, 85-168 Toruń, Poland; lukaszszukalski@gmail.com; 7Department of Hematology and Bone Marrow Transplantation, Pomeranian Hospitals in Gdynia, 81-519 Gdynia, Poland; judytastrzala87@gmail.com; 8Department of Hematology, St. John’s Cancer Center in Lublin, 20-090 Lublin, Poland; u.gosik@gmail.com; 9Department of Hematology, Provincial Specialist Hospital in Legnica, 59-220 Legnica, Poland; jmdebski@gmail.com; 10Individual Business Activity, ul. Zielna 28b/3, 51-313 Wroclaw, Poland; igaandrasiak@gmail.com; 11Harvard T.H. Chan School of Public Health—Executive and Continuing Education, Harvard University, Boston, MA 02115, USA; annaskotny@gmail.com

**Keywords:** CLL, pneumonia, venetoclax, overall survival, infection

## Abstract

This multicenter study assessed the incidence of pneumonia and its impact on survival in 322 patients with chronic lymphocytic leukemia (CLL) who were treated with venetoclax-based regimens. In real-world settings, patients with CLL who are treated with venetoclax-based regimens are at a higher risk of pneumonia, especially patients with R/R CLL who are treated with venetoclax in combination with rituximab compared to registration trials. Pneumonia has a negative impact on overall survival (OS) in patients with CLL who are receiving venetoclax-based therapy. However, in patients treated with first-line venetoclax and obinutuzumab, despite the high burden of comorbidities, the occurrence of pneumonia does not affect OS. Key risk factors for shorter event-free survival (EFS) in patients with CLL who were treated with venetoclax-based regimens included chronic obstructive pulmonary disease/asthma, spleen enlargement, elevated creatinine levels, and severe anemia; we identified a subgroup of patients at a high risk of pneumonia who particularly required anti-infective prophylaxis, appropriate treatment of underlying diseases, and close monitoring. Neutropenia, although common (59%), was not linked to an increased risk of pneumonia. These findings highlight the important issue of pneumonia and its impact on OS in the CLL patient population treated with venetoclax-based regimens, especially in patients following earlier lines of treatment, with R/R disease.

## 1. Introduction

Patients with chronic lymphocytic leukemia (CLL) are susceptible to infections that can affect clinical outcomes and contribute to a poor prognosis. The increased risk of severe infections is due to the underlying disease, comorbidities, and applied treatment.

Most infectious complications are associated with immunochemotherapy, which has a very limited role in the treatment of CLL in the current era [1].

In the present day, targeted therapies including Bruton’s kinase inhibitors (BTKi) and Bcl-2 inhibitors (venetoclax) constitute the standard of care for most patients with CLL, either as a first-line or subsequent treatment.

Fixed-duration therapy with venetoclax and an anti-CD20 monoclonal antibody, obinutuzumab, was associated with a 60% reduction in the risk of progression or death compared to treatment with chlorambucil and obinutuzumab. The 6-year follow-up of the CLL14 trial demonstrated the efficacy of this combination in untreated CLL patients regardless of TP53 or IGHV mutation status [2]. Similarly, in patients with relapsed/refractory (R/R) CLL, combination treatment with venetoclax and rituximab had a high response rate, including those with unfavorable genetic patterns such as 17p deletion [3]. Therefore, venetoclax-based regimens are widely used in the treatment of CLL and, as a fixed-duration therapy, are an attractive option for both first and subsequent lines.

Both the CLL-14 trial and the MURANO trial have shown that venetoclax treatment has an acceptable safety profile [4,5].

In the MURANO trial, pneumonia was observed in 9.3% of patients treated with rituximab plus venetoclax (R-VEN). In the CLL-14 trial, only 4.7% of patients treated with obinutuzumab plus venetoclax (O-VEN) had pneumonia. According to the European Society for Medical Oncology (ESMO)’s guidelines, antibiotic and antiviral prophylaxis is not recommended as routine practice for all treated patients with CLL. Only patients with recurrent infections or at a very high risk of developing infections should receive prophylaxis against pneumocystis pneumonia (PCP). Similarly, routine antifungal prophylaxis is not currently recommended. The National Comprehensive Cancer Network (NCCN)’s guidelines also indicate that prophylaxis is recommended only in selected groups of patients with CLL, treated mainly with PI3K inhibitors, chemoimmunotherapy with fludarabine or bendamustine, and alemtuzumab.

However, outside of registration trials, there are insufficient real-world data to estimate the incidence of infections that may affect treatment outcomes.

Pneumonia is one of the key serious adverse events that can have the greatest impact on prognosis.

Therefore, it seems reasonable to assess the occurrence of pneumonia and its predictive factors in the real world in CLL patients in the context of its impact on treatment outcomes and prognosis.

The aim of this study was to assess the incidence, impact on prognosis, and risk factors for pneumonia in CLL patients treated with venetoclax-based regimens at the centers of the Polish Adult Leukemia Study Group (PALG).

## 2. Methods

### 2.1. Study Population

This retrospective multicenter study was conducted in 8 hematological centers associated with the PALG. Data were collected between April 2019 and October 2023. For patients treated with O-VEN, it was first-line therapy. Patients treated with R-VEN constituted the group of patients with R/R CLL. R/R CLL patients were treated in the previous line with chemoimmunotherapy.

Researchers from each participating center provided data on consecutive patients with CLL treated with venetoclax-based regimens between 2019 and 2023. CLL diagnosis and eligibility for treatment were conducted by local researchers in accordance with current guidelines. The inclusion criteria included the following: age ≥ 18 years, a confirmed diagnosis of CLL with indications for treatment according to the guidelines of the International Workshop on Chronic Lymphocytic Leukemia (iwCLL) 2018 criteria, the Cumulative Illness Rating Scale (CIRS) > 6 points for newly diagnosed CLL patients treated with O-VEN, and R/R CLL after the ≥ 1st line of treatment (chemotherapy, immunochemotherapy) for patients treated with R-VEN. The exclusion criteria included the following: transformation of CLL to aggressive non-Hodgkin lymphoma (NHL) (Richter’s transformation) and a known history of infection with the human immunodeficiency virus (HIV).

The development of pneumonia was reported during treatment with venetoclax-based regimens in patients with CLL, and the definition was consistent with the National Cancer Institute Common Terminology Criteria for Adverse Events Assessment (CTCAE), version 5. Due to the retrospective nature of this study, only ≥3 grade pneumonia was reported. To eliminate collection bias, other pulmonary complications such as upper respiratory tract infections (sinusitis, tonsillitis, bronchitis) were not considered as differential factors due to the difficulty in defining them correctly based on retrospective analysis.

This study was performed in accordance with the Wroclaw Medical University Ethics Committee. Approval number: KB71/2024; date of approval: 27 June 2024.

### 2.2. Statistical Analysis

Numerical data were presented as median with interquartile range (IQR). Descriptive statistics were presented as absolute numbers with percentages. The K–S test was used to assess data distribution. A Mann‒Whitney U test was applied to compare quantitative data between 2 groups. Pearson’s chi-square analysis or Fisher’s exact test was used for intergroup comparison of categorical variables. Time-to-event data described in the manuscript included the following:(a)Overall survival (OS) was defined for patients without pneumonia as the time from the initiation of venetoclax therapy to death for any reason, whereas for patients with pneumonia it was defined as the time from pneumonia diagnosis to death for any reason.(b)Event-free survival (EFS) was defined as the time from the initiation of venetoclax therapy to the event—pneumonia or death from any reason, whichever occurred first.

OS was compared using a log-rank test between pneumonia and non-pneumonia patients. Odds ratio (OR) was used to assess the risk factors associated with the risk of developing the outcome of interest (independently of the time this happened), and hazard ratio (HR) was used to assess the risk factors associated with the time of outcome development. All ratios were presented with the corresponding 95% confidence intervals (CIs). Logistic regression and Cox regression models were applied for OR and risk factors identified with HR calculation, respectively. Univariable and multivariable analyses (MVA) were performed for all investigated patients and subgroups. Investigated risk factors included the following: sex, comorbidities, hypertension, chronic obstructive pulmonary disease (COPD), asthma, ischemic heart disease/heart failure (IHD/HF), chronic kidney disease, CIRS, presence of splenomegaly, B symptoms, neutropenia, anemia, creatinine level, hemoglobin level, white blood cell count, platelet count, lactate dehydrogenase (LDH), the CD4/CD8 lymphocyte ratio, β2-microglobulin level, and immunoglobulin IgG level. Variables initially selected based on their associations with increased risk of pneumonia in univariate analysis (*p* < 0.10; univariate analysis) were included for a stepwise regression model.

If a result was missing, the patient’s data were excluded from this particular analysis. All calculations were performed for the whole cohort of CLL patients treated with venetoclax-based regimens. Some additional calculations were performed separately—(1) for CLL patients treated with the R-VEN and (2) for CLL patients treated with the O-VEN.

Manuscript preparation was supported during the Harvard Medical School’s Clinical Scholars Research Training Program. The Polish Agency for Medical Research facilitated program participation.

## 3. Results

### 3.1. Baseline Characteristics

A total of 322 patients with CLL who were treated with venetoclax-based regimens in 8 hematological centers were included in this study (Table 1). A comparison of pneumonia incidence across registrational trials and real-world data are included in Appendix A).

Among these, 214 patients were treated with a rituximab–venetoclax regimen as a subsequent treatment line (R/R CLL), while 82 patients were treated with an obinutuzumab–venetoclax regimen as first-line CLL treatment. A total of 26 patients received venetoclax in combination with other agents (including BTKi)—not included in the analysis. The median age upon initiation of venetoclax treatment was 63 years; patients treated with O-VEN were older, with a median age of 68 years (*p* < 0.001). Most patients were male (62%). The most common comorbidity was hypertension (52%), with a slight predominance in patients treated with O-VEN (*p* = 0.05). Patients treated with the O-VEN regimen had more comorbidities, with a median CIRS score of 7 compared to a median CIRS score of 4 in patients treated with the R-VEN regimen, (*p* < 0.001) which reflects the inclusion criteria valid at the time of enrollment. The most common treatment complication was neutropenia (59%), occurring more frequently in patients treated with the R-VEN regimen (68%) compared to 46% in patients during the O-VEN therapy, (*p* = 0.001). The second most common complications of treatment were upper respiratory tract infections and pneumonia. During treatment with venetoclax-based regimens, 66 (20%) of patients developed pneumonia—50 (23%) patients in the R-VEN group and 13 (16%) patients in the O-VEN group (*p* = 0.15). Viral etiology of pneumonia was confirmed in 39 (59%) patients. In two patients, the etiology of pneumonia was unknown. The incidence of upper respiratory tract infections was similar among patients treated with the R-VEN and O-VEN regimens, both 32%. COVID-19 was diagnosed in 11% of all patients, with no significant differences between the treated patient cohorts. Patients treated with the R-VEN regimen compared to patients treated with the O-VEN regimen were characterized by significantly higher median LDH (259 vs. 216, *p* < 0.001), β2-microglobulin (4.4 vs. 3.2, *p* = 0.01), lower median immunoglobulin IgG levels (5.8 vs. 7.0, *p* = 0.009), and higher Endothelial Activation and Stress Index (EASIX) scores (2.3 vs. 1.6, *p* = 0.05). Deaths were reported in 47 (15%) patients—in 35 (18%) patients treated with the R-VEN regimen, in 5 (6%) patients treated with the O-VEN regimen, and in 7 (30%) treated with other venetoclax combination therapies.

The cumulative incidence of pneumonia was 20% in the entire cohort, with 23.5% and 15.9% in patients treated with R-VEN and O-VEN, respectively.

### 3.2. Clinical Characteristics of Venetoclax-Treated Patients with Pneumonia

Patients who developed pneumonia during venetoclax-based treatment were more likely to have COPD or asthma compared to patients without pneumonia (17% vs. 5%, *p* = 0.001). Patients with pneumonia compared to those without pneumonia had more comorbidities, with a median CIRS score of 6 vs. 4 (*p* = 0.04). Patients who developed pneumonia were characterized by a larger tumor mass indirectly expressed by an increased median diameter of the largest lymph node (4 cm vs. 3.3 cm, *p* = 0.04) and a higher incidence of splenomegaly (71% vs. 58%, *p* = 0.06). Patients with pneumonia were more frequently diagnosed with COVID-19 compared to patients without pneumonia (35% vs. 4%, *p* < 0.001). All of the results are presented in Table 2.

### 3.3. Prognostic Factors for Event-Free Survival (EFS) in All CLL Patients Treated with Venetoclax-Based Regimens

In univariate analysis, COPD/asthma, splenomegaly, anemia, and elevated levels of creatinine were associated with event occurrence (pneumonia or death, whichever occurred first) (HR = 3.19, 95%CI 1.71–5.94, *p* < 0.001; HR 2.11, 95%CI 1.2–3.72, *p* = 0.01; HR 4.18, 95%CI 2.45–7.12, *p* < 0.001; HR 3.11, 95%CI 1.44–6.69, *p* = 0.004, respectively). There was no association between neutropenia and IgG levels with pneumonia (Table 3).

In multivariate analysis, the aforementioned factors (COPD/asthma, splenomegaly, anemia, and elevated levels of creatinine) were confirmed as significant risk factors for EFS (HR = 2.08, 95%CI 1.16–3.74, *p* = 0.014; HR 1.73, 95%CI 1.08–2.78, *p* = 0.02; HR 3.58, 95%CI 2.18–5.89, *p* < 0.001; HR 2.13, 95%CI 1.10–4.11, *p* = 0.03, respectively) (Table 4).

### 3.4. Prognostic Factors for Event-Free Survival (EFS) in CLL Patients Treated with the R-VEN Regimen

In univariate analysis, the male sex, splenomegaly, anemia, and elevated levels of creatinine were risk factors for event occurrence (pneumonia or death) (HR 1.92, 95%CI 1.02–3.61, *p* = 0.044; HR 2.03, 95%CI 1.03–4.0, *p* = 0.042; HR 4.16, 95%CI 2.22–7.8, *p* < 0.001; HR 4.08, 95%CI 1.56–10.7, *p* = 0.004, respectively) (Table 5).

In the multivariate model, the male sex, anemia, and elevated levels of creatinine remained risk factors for EFS (HR 1.91, 95%CI 1.09–3.33, *p* = 0.02; HR 4.48, 95%CI 2.49–8.07, *p* < 0.001; HR 2.79, 95%CI 1.24–6.3, *p* = 0.01, respectively) (Table 6).

### 3.5. Overall Survival

The occurrence of pneumonia in CLL patients treated with venetoclax-based regimens was associated with inferior OS (*p* < 0.001) (Figure 1).

OS analyses were also performed for the two studied cohorts of patients (treated with the R-VEN and O-VEN regimen). Patients with R/R CLL treated with the R-VEN regimen, who developed pneumonia, had a worse OS (*p* < 0.001), whereas in a cohort of CLL patients treated with the O-VEN regimen, as a first-line therapy, the occurrence of pneumonia had no impact on OS (*p* = 0.45) (Figure 2 and Figure 3).

## 4. Discussion

Here, we present one of the largest real-world studies assessing the incidence and risk factors for pneumonia as one of the most serious and probably underestimated complications of treatment in patients with CLL who are treated with venetoclax-based regimens.

We showed that pneumonia is an unfavorable prognostic factor for OS in CLL patients treated with venetoclax-based regimens, especially in patients with R/R CLL who were treated with venetoclax in combination with rituximab. We found significantly higher rates of pneumonia in the entire cohort of CLL patients compared to the registration trials (the CLL-14 and the MURANO trial) [4,5]. This may be due to, on the one hand, the fact that most of the data were collected during the SARS-CoV-2 pandemic and, on the other hand, that pneumonia was more common in the cohort of R/R CLL patients who were intensively pretreated and had adverse clinical features.

In the MURANO trial, the incidence of pneumonia of any grade in R/R CLL patients treated with venetoclax plus rituximab was 9.3%, the most frequent type of infection [4].

Among CLL patients treated with venetoclax in the first line of the CLL-14 trial, the incidence of pneumonia, which, as in the MURANO trial, was the most common type of infection at grade ≥ 3, was 4.7% [5].

However, in a real-world study of the clinical efficacy and tolerability of venetoclax in combination with rituximab in patients with R/R CLL, the incidence of pneumonia was 23.9%, higher than in the MURANO registration trial [6].

The issue of infectious complications in CLL patients treated with venetoclax was analyzed in a real-world study conducted by the Italian Sorveglianza Epidemiologica Infezioni Fungine nelle Emopatie Maligne (SEIFEM) group [7]. The study showed the respiratory tract as the most commonly involved site of infections and singled out three specific variables (COPD, previous infections, and previous treatments) as risk factors for infections in CLL patients treated with venetoclax [7].

In our study, we confirmed respiratory tract infections as the most common infectious complication in CLL patients treated with venetoclax-based regimens.

We showed the occurrence of pneumonia in 20% of all CLL patients treated with venetoclax-based regimens. The incidence of pneumonia was clinically higher among patients treated with R-VEN compared to O-VEN, but without statistical significance. The viral etiology of pneumonia was clinically higher in patients treated with R-VEN.

These results differ from registration studies, which is obvious given the lack of selection of patients starting treatment in real-world settings. In our study, patients were generally older, with more comorbidities, and treated with more prior lines, reflecting the inclusion criteria for treatment with O-VEN and R-VEN regimens under the Ministry of Health program in Poland. In addition, the coincidence of the SARS-CoV-2 pandemic may have had an important influence on the high number of pneumonia cases observed in this study. Viral pneumonia was observed in the majority of all reported pneumonia (59%). According to ESMO and NCCN guidelines, antibiotic and antifungal prophylaxis is not recommended during venetoclax treatment. All included patients received antiviral prophylaxis (acyclovir), mainly due to combination treatment with monoclonal antibodies (rituximab or obinutuzumab).

The higher incidence of pneumonia in patients treated with R-VEN may be due to more advanced disease in patients with R/R CLL, treatment-related cumulative immunosuppression (longer drug-induced B-cell depletion, T cell immunodeficiency due to previous treatment), and immunosuppression due to progressive disease manifested mainly by reduced IgG and IgA levels [8]. Previous studies indicate that immunoglobulin replacement therapy may reduce the incidence of severe infectious complications [9,10].

Focusing on the potential predictors of pneumonia during venetoclax treatment, we found that the presence of COPD or asthma, splenomegaly, increased levels of creatinine at baseline, and anemia were the most predictive of pneumonia in all CLL patients.

In the most vulnerable R/R CLL patients treated with R-VEN, the key predictors of pneumonia were anemia and elevated creatinine levels.

Previous studies emphasized the significance of impaired renal function as a negative prognostic factor in patients with CLL [11]. In acquired pneumonia, renal insufficiency manifested by elevated blood urea nitrogen levels was associated with increased mortality in CLL patients [12].

We suggest that in the era of new targeted therapies, an increase in creatinine levels remains a significant negative prognostic factor. Our study showed that an increase in serum creatinine that does not meet the definition of acute renal failure can serve as a sensitive predictor of pneumonia during venetoclax treatment.

Splenomegaly is another risk factor for pneumonia observed in our study. In patients with more active disease expressed by the presence of an enlarged spleen, defenses against infectious agents may be weakened, increasing the risk of developing pneumonia during treatment. Previous data confirmed the potential associations between splenomegaly and the severity of lung involvement in patients with COVID-19 [13].

The presence of COPD is considered a risk factor for pneumonia [14,15]. We showed that underlying COPD/asthma is associated with pneumonia during treatment with venetoclax-based regimens. Our results are consistent with previously published real-world data from CLL patients treated with venetoclax, which identified COPD as a risk factor for severe infections [7,16].

A significant predictor of pneumonia in the entire cohort of CLL patients treated with venetoclax was severe anemia defined as hemoglobin below 8 g/dL. Anemia is a well-recognized adverse factor in patients with acquired pneumonia and, when severe, is independently associated with mortality [17]. Recently, the importance of the negative role of anemia has been emphasized in viral infections, including viral pneumonia. It has been shown that the presence of anemia was associated with disease progression and mortality in patients with both COVID-19- [18,19,20] and non-COVID-19-related pneumonia [17,21]. Therefore, given our results, it seems that hemoglobin level may be a simple, easily modifiable factor in predicting the risk of lung infection. The results of our study suggest that hemoglobin levels should be closely monitored in patients receiving venetoclax-based treatment, and perhaps normalizing hemoglobin levels may be the key to improving treatment outcomes.

Interestingly, we showed no association between the presence of neutropenia during treatment and pulmonary infections, suggesting that the most common complication of venetoclax treatment remains insignificant in terms of increasing the risk of infections.

Finally, our study showed that CLL patients treated with venetoclax-based regimens who developed pneumonia had worse OS than patients without pneumonia. This finding was particularly significant in a cohort of patients with R/R CLL who were treated with an R-VEN regimen. The lack of differences in OS between patients with and without pneumonia in the cohort of patients treated with O-VEN may be due to the first-line treatment and lower cumulative toxicity of previous therapies.

There were some limitations in our study. First, due to the retrospective nature of the study, complete data on all dates of pneumonia and the detailed microbiological etiology and antibiotic prophylaxis used were lacking. Second, we had limited data on vaccination history against SARS-CoV-2 and immune response, so this information as a potential confounding factor could not be included in the analysis.

## 5. Conclusions

In real-world settings, patients with CLL who are treated with venetoclax-based regimens are at a higher risk of pneumonia, especially patients with R/R CLL who are treated with venetoclax in combination with rituximab compared to registration trials.

Pneumonia has a negative impact on OS in patients with CLL receiving venetoclax-based therapy. In patients treated with first-line venetoclax and obinutuzumab, despite the high burden of comorbidities, the occurrence of pneumonia does not affect OS.

The presence of COPD/asthma, splenomegaly, severe anemia, and elevated creatinine levels are the key predictors/risk factors of pneumonia during treatment with venetoclax-based regimens, and we identified a subgroup of patients at a high risk of pneumonia who particularly required anti-infective prophylaxis, appropriate treatment of underlying conditions, and close monitoring. Neutropenia, as the most common hematological complication during venetoclax treatment, is not a risk factor for pneumonia.

In conclusion, our real-world study highlighted the important issue of pneumonia and its impact on OS in the CLL patient population treated with venetoclax-based regimens, especially in patients following earlier lines of treatment, with R/R disease.

## Figures and Tables

**Figure 1 cancers-16-04168-f001:** Kaplan–Meier survival curves for overall survival in all CLL patients treated with venetoclax-based regimens according to the occurrence of pneumonia. Median OS for patients without pneumonia was not reached. Median OS for patients with pneumonia was 12.6 (1.0-NA) months. Test log-rank *p* < 0.001.

**Figure 2 cancers-16-04168-f002:** Kaplan–Meier survival curves for overall survival in CLL patients treated with R-VEN regimen according to the occurrence of pneumonia. Median OS for CLL patients without pneumonia treated with the R-VEN regimen was not reached. Median OS for CLL patients with pneumonia treated with the R-VEN regimen was 12.5 (0.9-NA) months. Test log-rank *p* < 0.001.

**Figure 3 cancers-16-04168-f003:** Kaplan–Meier survival curves for overall survival in CLL patients treated with O-VEN regimen according to the occurrence of pneumonia. Median OS for CLL patients treated with O-VEN was not reached. Test log-rank *p* < 0.001.

**Table 1 cancers-16-04168-t001:** Clinical characteristics of patients with CLL (treated with R-VEN vs. O-VEN).

Variable	All Patients with CLL, n = 322	Patients with CLL Treated with R-VEN, n = 214	Patients with CLL Treated with O-VEN, n = 82	R-VEN vs. O-VEN, *p*
Age at start of VEN, median, IQR (years)	63 (56–70)	61 (55–69.0)	68 (61–73)	<0.001
Male, n (%)	201 (62)	132 (62)	55 (67)	0.389
**COMORBIDITIES**
Hypertension, n (%)	166 (52)	103 (48)	51 (62)	0.051
Diabetes mellitus, n (%)	63 (20)	34 (16)	24 (29)	0.013
COPD/asthma, n (%)	23 (7)	14 (7)	7 (9)	0.593
Stroke/TIA, n (%)	14 (4)	11 (5)	3 (4)	0.600
Supraventricular arrhythmia, n (%)	41 (13)	25 (12)	14 (17)	0.256
Coronary artery disease/stable heart failure, n (%)	43 (13)	24 (11)	18 (22)	0.023
Bowel disease, n (%)	27 (8)	23 (11)	4 (5)	0.103
Chronic kidney disease, n (%)	29 (9)	17 (8)	11 (13)	0.174
Venous thromboembolism, n (%)	16 (5)	10 (5)	6 (7)	0.404
Vascular disease, n (%)	54 (17)	29 (14)	22 (27)	0.009
Thyroid disease, n (%)	40 (12)	24 (11)	11 (13)	0.659
Other malignancies, n (%)	45 (14)	27 (13)	14 (17)	0.321
CIRS score	5.0 (2.0–8.0)	4.0 (2.0–8.0)	7.0 (4.0–9.0)	<0.001
CIRS > 6, n (%)	115 (36)	66 (31)	46 (56)	<0.001
**CLL characteristics**
Maximum nodal diameter, median, cm	3.5 (2.5–5.0)	3.7 (2.8–5.0)	3.5 (2.5–5.3)	0.953
Splenomegaly, n (%)	184 (57)	122 (62)	51 (62)	0.967
Autoimmune hemolysis, n (%)	33 (10)	24 (11)	4 (5)	0.104
Del 17p, n (%)	26 (8)	19 (10)	4 (8)	0.624
Del 11q, n (%)	63 (20)	49 (23)	9 (11)	
Lines of treatment before venetoclax, median	1 (0–2)	2 (1–3)	0 (0–0)	<0.001
**COMPLICATIONS**
Upper respiratory tract infection, n (%)	96 (30)	69 (32)	26 (32)	0.910
Pneumonia, n (%)	66 (20)	50 (23)	13 (16)	0.152
Viral etiology, n (%)	39 (59)	32 (64)	6 (46)	0.079
Non-viral etiology, n (%)	25 (38)	17 (34)	6 (46)	0.978
Neutropenia, n (%)	190 (59)	144 (68)	38 (46)	0.001
Anemia (Hb < 8 g/dL), n (%)	39 (12)	31 (15)	2 (2)	0.002
TLS, n (%)	40 (12)	13 (6)	9 (11)	0.150
Diarrhea/Intestinal infection, n (%)	24 (7)	16 (7)	6 (7)	0.963
Urinary tract infection, n (%)	4 (1)	2 (1)	1 (1)	1.000 *
Sepsis, n (%)	4 (1)	3 (1)	1 (1)	1.000 *
COVID-19, n (%)	34 (11)	23 (11)	7 (9)	0.573
**LABORATORY MEASURES at start of VEN**
White blood cells (G/L) × 10^9^/L; median, IQR	65.2 (20.1–146.2)	60.4 (20.1–144.2)	68.4 (22.6–136.9)	0.820
Lymphocytes (G/L) × 10^9^/L; median, IQR	32.8 (5.7–98.3)	34.7 (5.5–110.7)	33.5 (8.0–86.9)	
Hemoglobin (g/dL) median, IQR	11.3 (9.5–12.9)	11.4 (9.6–12.9)	11.9 (9.7–13.0)	0.257
Platelets (G/L) × 10^9^/L; median, IQR	124 (81–165)	120 (77–163)	132 (89–175)	0.102
Creatinine (mg/dL); median, IQR	0.91 (0.80–1.10)	0.92 (0.81–1.13)	0.94 (0.77–1.10)	0.743
LDH (U/L); median, IQR	244 (201–320)	259 (218–322)	216 (181–274)	<0.001
EASIX score; median, IQR	2.0(1.3–3.9)	2.3 (1.4–4.2)	1.61 (1.1–3.2)	0.050
Lymphocytes T CD4/C8 ratio; median, IQR	1.22 (0.92–1.76)	1.23 (0.94–1.58)	1.46 (1.0–2.0)	0.224
β2-microglobulin (mg/L); median, IQR	4.0 (2.8–5.4)	4.4 (2.9–5.9)	3.2 (2.7–4.1)	0.01
Immunoglobulin IgG (g/L); median, IQR	6.1 (4.4–8.3)	5.8 (4.1–7.9)	7.0 (5.0–9.4)	0.009
Clinical outcome, death, n (%)	47 (15)	35 (18)	5 (6)	0.012

* exact Fisher test.

**Table 2 cancers-16-04168-t002:** Clinical characteristics of patients with CLL treated with venetoclax-based regimens according to the occurrence of pneumonia.

Variable	CLL Patients with Pneumonia, n = 66	CLL Patients without Pneumonia, n = 253	*p*
Age at start of VEN, median, IQR (years)	61 (53–69)	63 (56–70)	0.371
Male, n (%)	45 (68)	154 (61)	0.275
**COMORBIDITIES**
Hypertension, n (%)	34 (52)	131 (53)	0.826
Diabetes mellitus, n (%)	12 (18)	49 (20)	0.763
COPD/asthma, n (%)	11 (17)	12 (5)	0.001
Stroke/TIA, n (%)	3 (5)	11 (4)	0.974
Supraventricular arrhythmia, n (%)	8 (12)	33 (13)	0.791
Coronary artery disease/stable heart failure, n (%)	10 (15)	33 (13)	0.707
Bowel disease, n (%)	9 (14)	18 (7)	0.103
Chronic kidney disease, n (%)	6 (9)	23 (9)	0.956
Venous thromboembolism, n (%)	5 (8)	11 (4)	0.292
Vascular disease, n (%)	14 (21)	40 (16)	0.338
Thyroid disease, n (%)	7 (11)	33 (13)	0.552
Other malignancies, n (%)	12 (18)	33 (13)	0.286
CIRS score	6.0 (3.0–9.0)	4.0 (2.0–8.0)	0.037
CIRS > 6, n (%)	29 (44)	86 (34)	0.146
**CLL characteristics**
Maximum nodal diameter, median, cm	4.0 (3.0–5.0)	3.35 (2.5–5.0)	0.039
Splenomegaly, n (%)	44 (71)	140 (58)	0.064
Autoimmune hemolysis, n (%)	5 (8)	27 (11)	0.448
Del 17p, n (%)	7 (14)	18 (9)	0.303
Lines of treatment before venetoclax, median	2 (2–3)	2 (1–3)	0.092
**COMPLICATIONS**
Neutropenia, n (%)	47 (72)	143 (57)	0.021
Anemia (Hb < 8 g/dL), n (%)	16 (25)	23 (9)	0.001
Sepsis, n (%)	1 (2)	3 (1)	1.000 *
COVID-19, n (%)	23 (35)	11 (4)	<0.001
Upper respiratory tract infection, n (%)	32 (48)	64 (25)	<0.001
Diarrhea/Intestinal infection, n (%)	7 (11)	17 (7)	0.286
Urinary tract infection, n (%)	1 (2)	3 (1)	1.000 *
**LABORATORY MEASURES**
White blood cells (G/L) × 10^9^/L; median, IQR	59.9 (20.1–139.8)	69.9 (21.9–150.7)	0.686
Hemoglobin (g/dL); median, IQR	11.2 (9.5–12.5)	11.5 (9.5–13.0)	0.216
Platelets (G/L) × 10^9^/L; median, IQR	114 (87–171)	125 (80–164)	0.652
Creatinine (mg/dL); median, IQR	0.96 (0.78–1.13)	0.91 (0.80–1.10)	0.653
LDH (U/L); median, IQR	243.5 (200–293)	244 (201.5–325)	0.613
EASIX score; median, IQR	1.75 (1.08–3.23)	2.20 (1.36–4.17)	0.141
CD4/CD8 ratio; median, IQR	2.4 (2.2–2.7)	1.2 (0.9–1.6)	0.073
Immunoglobulin IgG (g/L) median, IQR	6.3 (4.5–7.6)	6.1 (4.4–8.6)	0.724
Clinical outcome, death, n (%)	23 (35)	25 (11)	<0.001

* Fisher exact test was performed.

**Table 3 cancers-16-04168-t003:** Univariate analyses of EFS in CLL patients treated with venetoclax-based regimens.

Risk Factor	Category	HR	95% CI	*p* Value	OR	95% CI	*p* Value
Sex	Female	0.76	(0.47–1.25)	0.28	0.81	(0.48–1.34)	0.41
Hypertension	Yes	1.04	(0.65–1.65)	0.88	1.07	(0.65–1.74)	0.80
COPD/Asthma	Yes	3.19	(1.71–5.94)	<0.001	4.3	(1.78–10.37)	0.001
Coronary artery disease/stable heart failure	Yes	1.41	(0.76–2.63)	0.27	1.37	(0.69–2.71)	0.37
Chronic kidney disease	Yes	1.46	(0.70–3.05)	0.32	1.32	(0.59–2.97)	0.50
CIRS	>6	1.38	(0.86–2.22)	0.18	1.58	(0.96–2.6)	0.07
Splenomegaly	Yes	2.11	(1.2–3.72)	0.01	2.12	(1.22–3.68)	0.007
B symptoms	Yes	1.32	(0.76–2.31)	0.33	1.22	(0.7–2.13)	0.48
Neutropenia (ANC < 1.0 G/L)	Yes	1.04	(0.64–1.69)	0.87	1.51	(0.9–2.52)	0.11
Anemia (Hb < 8 g/dL)	Yes	4.18	(2.45–7.12)	<0.001	4.02	(2.0–8.02)	<0.001
EASIX score	No	0.99	(0.94–1.04)	0.72	0.97	(0.91–1.02)	0.23
Creatinine (mg/dL)	No	3.11	(1.44–6.69)	0.004	2.62	(1.05–6.57)	0.039
Hemoglobin (g/L)	No	0.87	(0.79–0.96)	0.008	0.9	(0.81–1.01)	0.058
White blood cells (G/L)	No	1.0	(1.0–1.0)	0.07	--	---	---
Platelets (G/L)	No	1.0	(1.0–1.0)	0.90	--	---	---
LDH (U/L)	No	1.0	(1.0–1.0)	0.86	--	---	---
Lymphocytes CD4/CD8 ratio	No	1.17	(0.84–1.64)	0.35	1.26	(0.81–1.97)	0.30
B2-microglobulin (mg/L)	No	1.01	(0.89–1.15)	0.88	1.03	(0.89–1.19)	0.65
Immunoglobulin IgG (g/L)	No	0.93	(0.83–1.04)	0.19	0.94	(0.84–1.01)	0.23

**Table 4 cancers-16-04168-t004:** Multivariate analyses of EFS in CLL patients treated with venetoclax-based regimens.

Risk Factor	Category	HR	95% CI	*p* Value
COPD/Asthma	Yes	2.08	(1.16–3.74)	0.014
Splenomegaly	Yes	1.73	(1.08–2.78)	0.02
Anemia (Hb < 8 g/dL)	Yes	3.58	(2.18–5.89)	<0.001
Creatinine (mg/dL)	No	2.13	(1.10–4.11)	0.03

Cox regression model, Chi2 = 39.68; df = 4; *p* < 0.001.

**Table 5 cancers-16-04168-t005:** Univariate analyses of EFS in CLL patients treated with R-VEN regimen.

Risk Factor	Category	HR	95% CI	*p* Value	OR	95% CI	*p* Value
Sex	FemaleMale	0.521.92	(0.28–0.98)(1.02–3.61)	0.044	0.551.8	(0.28–0.98)(0.3–1.04)	0.064
Hypertension	Yes	0.93	(0.53–1.62)	0.8	1.01	(0.55–1.86)	0.963
COPD/Asthma	Yes	2.09	(0.89–4.91)	0.092	1.66	(0.55–5.03)	0.366
Coronary artery disease/stable heart failure	Yes	1.25	(0.56–2.78)	0.583	1.33	(0.55–3.23)	0.531
Chronic kidney disease	Yes	1.5	(0.6–3.8)	0.387	1.55	(0.56–4.31)	0.394
CIRS	>6	1.57	(0.89–2.76)	0.119	1.88	(1.02–3.49)	0.043
Splenomegaly	Yes	2.03	(1.03–4.0)	0.042	2.08	(1.07–4.06)	0.031
B symptoms	Yes	1.13	(0.61–2.11)	0.690	1.1	(0.58–2.07)	0.767
Neutropenia (ANC < 1.0 G/L)	Yes	1.03	(0.56–1.89)	0.923	1.35	(0.71–2.57)	0.364
Anemia (Hb < 8 g/dL)	Yes	4.16	(2.22–7.8)	<0.001	3.36	(1.53–7.38)	0.002
EASIX score		0.99	(0.93–1.06)	0.781	0.95	(0.88–1.03)	0.248
Creatinine (mg/dL)		4.08	(1.56–10.7)	0.004	2.77	(0.9–8.53)	0.073
Hemoglobin (g/L)		0.88	(0.78–0.98)	0.026	0.92	(0.81–1.04)	0.174
White blood cells (G/L)		1.0	(1.0–1.0)	0.153	--	---	---
Platelets (G/L)		1.0	(1.0–1.01)	0.433	--	---	---
LDH (U/L)		1.0	(1.0–1.0)	0.725	--	---	---
Lymphocytes CD4/CD8 ratio		1.07	(0.58–1.98)	0.818	1.09	(0.54–2.22)	0.795
B2-microglobulin (mg/L)		0.99	(0.83–1.19)	0.955	0.99	(0.81–1.21)	0.9
Immunoglobulin IgG (g/L)		0.90	(0.77–1.04)	0.148	0.91	(0.79–1.05)	0.192

**Table 6 cancers-16-04168-t006:** Multivariate analyses of EFS in CLL patients treated with R-VEN regimen.

Risk Factor	Category	HR	95% CI	*p* Value
Male sex	Yes	1.91	(1.09–3.33)	0.02
Anemia (Hb < 8 g/dL)	Yes	4.48	(2.49–8.07)	<0.001
Creatinine (mg/dL)	No	2.79	(1.24–6.30)	0.01

Cox regression model, Chi2 = 27.8; df = 3; *p* < 0.001.

## Data Availability

The datasets generated and/or analyzed during this current study are available from the corresponding author on reasonable request.

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
