# Peer review of "RETRACTED: Pneumonia in Patients with Chronic Lymphocytic Leukemia Treated with Venetoclax-Based Regimens: A Real-World Analysis of the Polish Adult Leukemia Group (PALG)"

_cancers, 2024, doi:10.3390/cancers16244168_

Round 1
Reviewer 1 Report
Comments and Suggestions for Authors
- Vaccination status (covid, influenza and pneumococcal vaccination) should be reported, if available.
- Row 191 – “26 patients received venetoclax in 191 combination with other agents (including BTKi) – not included in the analysis”
This could be an interesting aspect (maybe in future analysis) to investigate if combining venetoclax with antiCD20 or with BTKi could lead to different risk of developing pneumonia in a real life setting. Data from CLL14 and Glow trial show a similar rate of pneumonia (5.7% in the Glow trial), but it could be different in a real life population.
- Row 339 “immunosuppression due to progressive disease manifested mainly by reduced IgG and IgA levels.”
IgG levels and possibile Ig replacement therapy should be reported, if available. It would be interesting to investigate if hypogammaglobulinemia is a risk factor for developing pneumonia in this population. A study from Germany reported that the introduction of immunoglobulin replacement reduced the risk of severe infectious complications in hematologic patients. See “Link H et al , Immunoglobulin substitution in patients with secondary antibody deficiency in chronic lymphocytic leukemia and multiple myeloma: a representative analysis of guideline adherence and infections. Support Care Cancer 2022;30:5187–200. https://doi.org/10.1007/s00520-022-06920-y”
More recently Mikulska et al. recommended immunoglobulin replacement in case of hypogammaglobulinemia and recurrent or severe infections, particularly respiratory. See Mikulska et al., Prevention and management of infectious complications in patients with chronic lymphocytic leukemia (CLL) treated with BTK and BCL-2 inhibitors, focus on current guidelines Blood Reviews, https://doi.org/10.1016/j.blre.2024.101180
Author Response
Reviewer 1:
- Vaccination status (covid, influenza and pneumococcal vaccination) should be reported, if available.
Answer 1: We agree, The limitation of the study included information on the lack of availability of reliable data on vaccinations performed as well as immune response due to the retrospective character of the study.
- Row 191 – “26 patients received venetoclax in 191 combination with other agents (including BTKi) – not included in the analysis”. This could be an interesting aspect (maybe in future analysis) to investigate if combining venetoclax with antiCD20 or with BTKi could lead to different risk of developing pneumonia in a real life setting. Data from CLL14 and Glow trial show a similar rate of pneumonia (5.7% in the Glow trial), but it could be different in a real life population.
Answer 2: Thank you for this comment. In the analysis presented here, 26 patients received venetoclax in combination with other drugs (including BTKi inhibitors). Because the number of patients receiving such combinations was too small, the sample was not representative enough to be included in the statistical analysis. We plan to expand the analysis in future publications to include patients receiving a combination treatment with Bcl-2 inhibitor a BTK inhibitor.
- Row 339 “immunosuppression due to progressive disease manifested mainly by reduced IgG and IgA levels.”
IgG levels and possibile Ig replacement therapy should be reported, if available. It would be interesting to investigate if hypogammaglobulinemia is a risk factor for developing pneumonia in this population. A study from Germany reported that the introduction of immunoglobulin replacement reduced the risk of severe infectious complications in hematologic patients. See “Link H et al , Immunoglobulin substitution in patients with secondary antibody deficiency in chronic lymphocytic leukemia and multiple myeloma: a representative analysis of guideline adherence and infections. Support Care Cancer 2022;30:5187–200. https://doi.org/10.1007/s00520-022-06920-y”
More recently Mikulska et al. recommended immunoglobulin replacement in case of hypogammaglobulinemia and recurrent or severe infections, particularly respiratory. See Mikulska et al., Prevention and management of infectious complications in patients with chronic lymphocytic leukemia (CLL) treated with BTK and BCL-2 inhibitors, focus on current guidelines Blood Reviews, https://doi.org/10.1016/j.blre.2024.101180
Answer 3: Thank you for your valuable remark. In our analysis, IgG immunoglobulin levels were not determined in all studied patients. However, the available data did not indicate a significant immunoglobulin deficiency in the analyzed group of patients. Still, incomplete data obtained due to the retrospective analysis do not allow us to conclude whether the deficiency or lowered level of immunoglobulins impacts infectious complications in this group of patients. This is undoubtedly a significant clinical problem that requires deeper analysis in prospective studies. We have added this valid point to the discussion and cited the indicated publications (lines 341-342).
Reviewer 2 Report
Comments and Suggestions for Authors
(1) I understand that the overall risk factors for pneumonia in CLL are chronic obstructive pulmonary disease/asthma, splenomegaly, elevated creatinine levels, and severe anemia. Does this include maximum lymph node diameter? (2) It would be better to conduct sub-analysis by dividing the patients into 1) newly diagnosed CLL group and r/rCLL group, 2) COVID19-infected group and COVID19-uninfected group, and 3) groups with initial regimen R-VEN and O-VEN. It would be easier to understand if there was a summary table of these. (3) Neutropenia is not a risk factor for pneumonia. Is the use of G-CSF the cause? (4) I understand that this variable was initially selected based on its association with increased pneumonia risk in univariate analysis (p<0.10, univariate analysis), but I think P<0.05 would be preferable. (5) It would be easier to understand if there was a comparison table with other similar research reports. For example, it would be good if you could summarize the MURANO registration study, CLL-14 study, etc. (6) Please explain why the O-VEN group had a lower risk of pneumonia.
Author Response
Rewiever 2:
(1) I understand that the overall risk factors for pneumonia in CLL are chronic obstructive pulmonary disease/asthma, splenomegaly, elevated creatinine levels, and severe anemia. Does this include maximum lymph node diameter?
Answer 1: Maximum lymph node diameter remained statistically insignificant in the context of pneumonia incidence in both univariate and multivariate analysis.
(2) It would be better to conduct sub-analysis by dividing the patients into 1) newly diagnosed CLL group and r/rCLL group, 2) COVID19-infected group and COVID19-uninfected group, and 3) groups with initial regimen R-VEN and O-VEN. It would be easier to understand if there was a summary table of these
Answer 2: Our goal was not to compare patients who developed COVID-19 with patients who did not, therefore such a comparison was not included in the manuscript. The presented division of patients into R-VEN and O-VEN regimens corresponds to patients with refractory/relapsed disease treated with R-VEN (in the 2nd or subsequent line of treatment) and patients treated in the first line with O-VEN regimen, respectively.
In Poland, the O-VEN regimen is a treatment used in the first line of therapy, while the R-VEN regimen is a treatment used in patients with refractory/relapsed CLL, in 2 or more lines of treatment.
Qualification for each type of treatment (R-VEN and O-VEN) is described in the methods section (lines 111-114).
(3) Neutropenia is not a risk factor for pneumonia. Is the use of G-CSF the cause?
Answer 3: In univariate analysis, we considered only baseline neutrophil counts and analyzed their association with the occurrence of pneumonia during venetoclax treatment. Our results indicate that baseline neutropenia (absolute neutrophil count below 1 G/L) is not associated with the occurrence of pneumonia during venetoclax therapy.
We did not analyze the association of neutropenia that occurred during venetoclax treatment (as an adverse event of therapy) on the occurrence of pneumonia.
(4) I understand that this variable was initially selected based on its association with increased pneumonia risk in univariate analysis (p<0.10, univariate analysis), but I think P<0.05 would be preferable
Answer 4:
We decided to use p < 0.10 instead of p < 0.05 to increase sensitivity, because using a higher threshold (p<0.10) helps detect more potential risk factors. The second reason is to include as many potentially important variables as possible in the multivariate analysis. Therefore, it is acceptable to adopt a less stringent threshold to ensure that important factors are not omitted.
(5) It would be easier to understand if there was a comparison table with other similar research reports. For example, it would be good if you could summarize the MURANO registration study, CLL-14 study, etc.
Answer 5: Thank you for this valuable remark. We added Supplementary Table 1 with a comparison of pneumonia incidence across registrational trials and our real-world study (lines 185-187).
6) Please explain why the O-VEN group had a lower risk of pneumonia.
Answer 6: Patients treated with the O-VEN regimen were not previously treated with any other therapy. O-VEN therapy was their first line of treatment. Therefore, this is a group of patients who are less overtreated, with a lower risk of immune system dysfunction, and potentially less prone to infections. However, this is just an assumption. The occurrence of the most serious complication, pneumonia, has not been studied in this group of patients in a real-world setting.
Round 2
Reviewer 2 Report
Comments and Suggestions for Authors
Authors' answers to my comments are acceptable.
Comments on the Quality of English LanguageEnough quality of English